# Psychological Profile in Women with Chronic Pelvic Pain

**DOI:** 10.3390/jcm11216345

**Published:** 2022-10-27

**Authors:** Mónica Magariños López, María José Lobato Rodríguez, Ángela Menéndez García, Sophie García-Cid, Ana Royuela, Augusto Pereira

**Affiliations:** 1Department of Psychiatry, Puerta de Hierro University Hospital, 28222 Madrid, Spain; 2Clinical Psychology, Puerta de Hierro University Hospital, 28222 Madrid, Spain; 3Biostatistics Unit, IDIPHISA, CIBERESP, Puerta de Hierro University Hospital, 28222 Madrid, Spain; 4Department of Gynecology and Obstetrics, Puerta de Hierro University Hospital, 28222 Madrid, Spain

**Keywords:** chronic pelvic pain, personality traits, depression, anxiety, catastrophizing, pain acceptance

## Abstract

(1) Background: Chronic Pelvic Pain (CPP) is a prevalent medical condition with a complex treatment due to different variables that influence its clinical course. (2) Methods: Psychological variables such as depression, anxiety, catastrophizing or neuroticism have been described as influencing CPP. This is a cross-sectional study of 63 patients with CPP sent for a psychological evaluation due to participation in group therapy for CPP. The main purpose of this study was to characterize the baseline psychological characteristics of women with CPP. The NEO Five Factor Inventory (NEO-FFI), State and Trait Anxiety Inventory (STAI), Beck Depression Inventory-Fast Screen (BDI-FS), Pain Catastrophizing Scale (PCS) and Chronic Pain Acceptance Questionnaire (CPAQ) were performed. (3) Results: The personality profile of patients (NEO FFI) shows high neuroticism, low extraversion and low conscientiousness. The 25.4% of patients had moderate or severe depression according to BDI-FS results, almost half of the patients had high levels of anxiety trait (>P75, 49.2%) and more than half the patients had high levels of anxiety state (>P75, 59.5%). Punctuations of PCS and CPAQ are similar to patients with fibromyalgia. (4) Conclusions: CPP is associated with high levels of depression, anxiety, neuroticism, catastrophizing and low pain acceptance. It is important to develop interventions that can modify these psychological factors in order to improve the clinical course of CPP.

## 1. Introduction

Chronic Pelvic Pain (CPP) is a debilitating condition that is present in 4–43.4% of women worldwide according to different studies [1,2]. In Spain, it afflicts 22.8% of the population, affecting 30.9% of woman and 15.6% of men [3]. CPP represents a considerable economic burden on women and healthcare systems worldwide, although we do not have exact figures on the cost of it [4].

CPP is a pain localized at the lower abdomen, pelvis or pelvic structures, that persists for at least six months and occurs continuously or intermittently [3]. It can arise from the gynecologic, urologic, gastrointestinal, musculoskeletal and neurovascular systems, and many factors contribute to its development. The most common gynecological causes of CPP are endometriosis, pelvic inflammatory disease and adhesions, the most common digestive cause is irritable bowel syndrome and the most frequent urologic causes are chronic prostatitis and interstitial cystitis [5,6]. The association of CPP with pain conditions in another location is not infrequent [5,7,8]. In no urologic CPP, it was found that 40% of patients had multiple pain syndromes [7], and in urologic CPP, 38% were associated to other pain syndromes [5]. Fibromyalgia, irritable bowel syndrome and chronic fatigue syndrome were those more frequently associated with CPP [5,7].

Some key factors in CPP such as pelvic pathology, history of abuse or comorbid psychological conditions like depression, anxiety or catastrophizing have been described [2,9].

Not many studies have focused on the psychological profiles of women with CPP [9,10,11,12]. A study by Bryant et al. found significant symptoms of depression and anxiety as measured by the Hospital Anxiety and Depression Scale (HADS). More than half of the patients (53.1%) in the experiment experienced moderate or severe anxiety symptoms, and 26.9% of the patients who reported depression had moderate or severe symptoms [9]. Anxiety and depression were found to be correlated in women with CPP [13], and catastrophizing was found to be positively correlated with depression scores [11].

The personality trait that has been most studied in chronic pain is neuroticism. A considerable number of studies have found higher levels of neuroticism compared to control groups in various chronic pain types [14,15,16,17,18,19,20,21,22,23,24,25]. Other studies suggest that neuroticism may be a causal factor for chronic pain [14,26,27,28] and much research has focused on neuroticism as a moderator of the pain experience [29], through the creation of a heightened physical sensitivity to pain due to its inherent connection with chronically moderated anxiety [14,22,30,31,32,33,34] or influencing adjustment to pain by affecting pain-related beliefs and emotional distress [14,19,35,36,37,38,39].

There is a paucity of data regarding the role of personality traits in CPP. One study investigated the association of personality traits with the baseline clinical characteristics and treatment outcomes of patients with chronic prostatitis/chronic pelvic pain syndrome (CP/CPPS), finding that neuroticism may be the most important personality trait associated with treatment response and the severity of depression and somatization in patients with CP/CPPS [40]. Another cross-sectional study [41] explored temperament and character dimensions in women with endometriosis and pelvic pain using the Temperament and Character Inventory-Revised (TCI-R). They found that higher harm avoidance and lower self-directedness were associated with a greater severity of chronic pelvic pain [41].

The main purpose of this study was to characterize the baseline psychological characteristics of women treated in a multidisciplinary group of CPP. The secondary goal was to investigate if psychological and personal variables are related to each other and in which way.

Given the importance of psychological and personality variables in chronic pain in general and in CPP in particular [2,5,6,7,8,9,10,11,12,13,14,15,16,17,18,19,20,21,22,23,24,25,26,27,28,29,30,31,32,33,34], the paucity of studies on women and the need to expand knowledge in this field, the aim of this work was to investigate whether these variables are also present in the CPP women patients of our study and to expand, if possible, the understanding we have about the psychological profile of these patients. We also wondered whether we could find personality traits other than neuroticism that would shape a personality profile.

Our main hypothesis is that the study participants will have high levels of depression, anxiety and catastrophizing, low acceptance of pain and high levels of neuroticism as part of the personality dimension. Our secondary hypothesis is that anxiety, depression and catastrophizing may be positively correlated.

## 2. Materials and Methods

### 2.1. Patients

This is a cross-sectional descriptive study without a comparison group. The participants were comprised of sixty-three women aged 18 and older with CPP. They were referred between 2019 and 2022 by professionals from urology, gynecology and digestive departments to the psychiatry department of the University Hospital Puerta de Hierro Majadahonda, Madrid (Spain) to be evaluated prior to entering a psychotherapeutic group for CPP. The criteria the deriving physicians used for referral to psychiatry evaluation were emotional distress and/or the presence of psychopathology associated with CPP that interfered with its outcome. All patients who agreed to fill in the questionnaires and that these could be used for research purposes were included. The verbal informed consent was obtained at the end of the evaluation visit. Participants did not receive any financial compensation. Ninety patients were referred for evaluation. Of these, five did not attend the scheduled appointment, and twenty-one did not participate in the research. However, non-participation in the research did not exclude the possibility of joining the psychotherapy group.

This research is part of a larger project that aims to assess the effectiveness of the psychotherapy group in CPP. The project of which this study is a part was approved by the Drug Research Ethics Committee.

### 2.2. Data Assessed

A clinical interview ad hoc was performed by a clinical psychologist or a psychiatrist, who are part of the Multidisciplinary Group for CPP Management. The instruments below were used for the assessment of the patients.

#### 2.2.1. Clinical Interview

An unstructured interview was carried out to collect sociobiographical data as well as the most relevant clinical history. Before the clinical interview, the patient’s electronic medical record was reviewed in order to obtain the most precise history as possible. Both psychiatric and history of pain variables (previous psychiatric diagnosis and treatments, years of evolution of pain, pain treatments tried, activities that improved and worsened pain, degree of affectation and perceived support) were explored. Although the evaluators had a script to cover all the relevant information that we wanted to assess, this was not always possible, which explains the loss of some data.

#### 2.2.2. NEO Five-Factor Inventory (NEO-FFI)

This is the validated Spanish version of the NEO-FFI, a scale of 60 items selected from the original Revised NEO Personality Inventory (NEO-PI-R) to carry out a rapid personality assessment. The inventory provides a rapid personality configuration based on the five major factors: extraversion (E), neuroticism (N), agreeableness (A), openness (O) and conscientiousness (C). Each of these factors is assessed by 12 Likert-type items to be answered on a scale from Strongly Disagree to Strongly Agree [42].

#### 2.2.3. Beck Depression Inventory Fast Screen (BDI-FS)

This scale is an abbreviated adaptation of the Beck Depression Scale (BDI) with only 7 items with the aim of screening for possible depressive symptomatology [43].

#### 2.2.4. State Trait Anxiety Inventory (STAI)

This is one of the most widely used tests in clinical settings for the assessment of anxiety since, in a brief form, it evaluates both current anxiety symptoms (SA) and anxiety as a stable trait (TA) of the subject’s personality. It consists of 40 Likert-type items ranging from Almost Never (0) to Almost Always (3) [44].

#### 2.2.5. Chronic Pain Acceptance Questionnaire (CPAQ)

This questionnaire is based on the assumption that acceptance rather than avoidance in chronic pain patients is a predisposing factor for improved daily functioning. It is composed of 20 Likert-formatted items and assesses both pain willingness (PW) as well as the subject’s willingness to engage in activities (AE). It has been widely reviewed and validated. In Spain it has been validated in fibromyalgia, a pathology that causes chronic pain [45,46]. Higher punctuations indicate higher acceptance of pain. Maximum total punctuation is 120, being PW a maximum of 54 and AE a maximum of 66.

#### 2.2.6. Pain Catastrophizing Scale (PCS)

This is a Spanish adaptation of the PCS, given the weight that this psychological variable has in the course of CPP; we considered it necessary to include an instrument that would specifically assess it and, later on, check whether the treatment proposed would have an impact on it. The Spanish version of this questionnaire is made up of 13 statements to which the patient has to give a score from 0 (Not at all) to 4 (All the time) according to his or her degree of agreement with the statement The PCS yields a total score ranging from 0 to 52 and has three subscales assessing rumination, magnification, and helplessness. It has been used and validated in different types of pain, including endometriosis [47,48].

### 2.3. Data Analysis

Descriptive statistics were mainly extracted for the means and 25, 50 and 75 percentiles of the assessment scales. A one-sample Student’s t-test was performed to compare the means of CPAQ and PCS of our sample group with the means provided by the Spanish validation of both scales, for the population of fibromyalgia and endometriosis, respectively. Subsequently, Pearson correlations analyses between the different scales and subscales with each other were carried out. *p* value < 0.05 was considered statistically significant. Statistical software Stata v. 16 was used for data analysis.

## 3. Results

### 3.1. Sociodemographic and Clinical History Results

All the participants were women aged between 22 and 79 years, with a mean age of 44 and a standard deviation (SD) of 12.6 years. Most of the patients came from the gynecology department; 68.2% of them were married or in a couple, 7.9% were divorced and 23.8% were singles; 59.7% had children and 40.5% did not; 42.8% had university studies; 30.2% were active workers, 20.6% were unemployed and 28.6% had a work incapacity (Table 1).

A substantial number of women had a past psychiatric history (73.2%), depression being the most prevalent of the psychiatric disorders (21.4%), followed by anxiety disorders (14.3%) and adaptative disorders (12.5%). Only 26.8% of patients had no past psychiatric history. Similarly, 63.3% of patients had a family psychiatric history, depression being the most prevalent disorder (30.6%), followed by substance use disorder (8.2%) or psychosis (8.2%). Most of the patients had current pain treatment (80%) and 44.2% had psychopharmacological treatment. The substance consumed most frequently was alcohol (35.9%) followed by tobacco (23.7%). Only 5.1% consumed cannabis and 2.6% cocaine. The data regarding past history abuse show that 41.18% had past abuse (Table 1).

### 3.2. Clinical Scales Results

In the personality scale NEO-FFI, we found high mean (P25–P75) levels of neuroticism (63.2, 56–71), low extraversion (42.4, 34–52) and low conscientiousness (42.9, 34–51) (Table 2). It should be noted that, with respect to the NEO-FFI scale, we show the transformed scores as well as, in parentheses, the range to which this score is equivalent, according to the following measure: Very low (VL) 25–34; Low (L) 35–44; Medium (M) 45–54; High (H) 55–64 and Very high (VH) 65–75.

The data obtained from the BDI-FS are shown in Table 3. This scale gives each score a range of clinical severity from minimal to severe. Following the same dynamic, we present the sample mean obtained in the BDI-FS and the range to which it corresponds, as well as the sample percentages obtained in each of the correction ranges. It can be observed that almost 74.6% of the sample indicates no depressive symptoms or a mild level, while 25.4% identify medium and severe depressive symptomatology.

The State and Trait Anxiety Scale (STAI) results are shown in Table 4. Initially, only the Trait Anxiety subscale was administered in the clinical assessment, later including State Anxiety as well. This change explains the difference in sample size at both levels.

The STAI correction offers, for each score, the percentile to which it corresponds according to its normative group, in this case, the general population of adult women. It can be seen in the table that more than 70% of our sample obtains scores above the 50th percentile (78.6% in the case of State Anxiety (SA) and 71.2% in Trait Anxiety (TA), and in both cases, most of this sample is between the 85 and 99th percentile (38.1% and 27.1%). The sample mean corresponds to a 75–85 percentile in the case of SA and a 50–75 percentile in TA.

The results of the Pain Catastrophizing Scale (PCS) and Chronic Pain Acceptance Questionnaire (CPAQ) are shown in Table 2. The results show that there was no significant difference between our CPP sample and the means in the population diagnosed with fibromyalgia with the CPAQ scale (T = −0.59 *p* = 0.55). There was also no significant difference between our sample mean and that provided by the PCS scale for endometriosis (T = 0.79 *p* = 0.42).

### 3.3. Correlation Analysis between Different Measures

The correlation analyses are presented indicating with * those clinically relevant (r of Pearson > ±0.6). Correlations of levels of the same variable are not indicated. Table 5 shows the correlations between the different clinical scales, while Table 6 shows the correlations obtained between the clinical scales and the NEO-FFI.

Regarding the STAI scale, Table 5 shows that both subscales, STAI-SA and STAI-TA, correlate positively and significantly with the BDI, although this correlation is higher in the case of STAI-TA (r of Pearson = 0.74) than in STAI-SA (r of Pearson = 0.64). The same does not occur in the correlation with the PCS, in which the relationship is higher with STAI-TA than with STAI-SA (r of Pearson = 0.53 vs. r of Pearson = 0.30) and not clinically relevant.

The overall analysis of the CPAQ does not yield clinically significant correlations, although it seems important to point out that in its relationship with the STAI-SA, STAI-TA and BDI, most of the inverse correlation found is explained by the activity index (r of Pearson = −0.59, r of Pearson = −0.57 and r of Pearson = −0.57 respectively) and not by the pain acceptance subscale (r of Pearson = −0.16, r of Pearson = −0.21, r of Pearson = −0.22). 

Finally, with respect to the relationship between PCS and CPAQ, although no correlation could be considered relevant, it can be noted that the Helplessness subscale of PCS is the one with the highest correlation with CPAQ (r of Pearson = −0.45).

The analysis of correlations of the clinical variables with the personality assessment subscales did not yield relevant correlations (Table 6). Only a clinically significant positive correlation was found between the variable Neuroticism and STAI-TA (r of Pearson = 0.65), higher than that obtained with STAI-SA (r of Pearson = 0.53).

## 4. Discussion

Chronic pelvic pain is a clinical condition that affects a significant percentage of the population, especially women (almost twice as common in women as it is in men) [3,49,50]; in fact all of our patients were women.

Our results confirm our main hypothesis. We have found that 57.6% of the women have depressive symptoms with a 25.4% of the sample having medium or severe symptoms. More than 70% of the patients have high levels of anxiety obtaining scores above the 50th percentile on the STAI scale (78.6% STAI-SA and 71.2% STAI-TA). Catastrophizing can be considered high with half of the patients with scores of 29 and above (Maximum of the scale 52 points. P50 = 29 and P75 = 38) and low illness acceptance (Maximum of the scale 120. P50 = 49 and P75 = 58). Regarding personality traits, we found high neuroticism as hypothesized, but we also found low extraversion and low conscientiousness, a finding not described as a cluster in previous studies in this population.

Regarding the secondary hypothesis, we found positive relevant clinical correlations between anxiety and depression (r of Pearson = 0.74 for STAI-TA and 0.64 for STA-SA) as hypothesized but did not find correlations of anxiety and depression with catastrophizing. Unexpectedly, we found a relevant correlation between neuroticism and trait anxiety (r of Pearson 0.65). 

Below we discuss in detail the different results obtained in our study.

With regard to sociodemographical data, we found that almost half of the patients do not work either because they are unemployed (20.6%), on temporary leave (27.3%) or on permanent incapacity for work (5.5%). Data of unemployment in Spanish women (over 25 years of age) between 2019 and 2022 ranged between 14–16.9% [51], being this parameter higher in our sample. One study by Haugstad et al. in women with CPP found a bit higher levels of temporary leave (14 of 40 patients −35%) and disabled patients (5 of 40 patients −12.5%) [52]. Previous studies that analyzed the costs associated to different pathologies that produce CPP find that productivity costs are the greatest contributors to overall costs [4,53,54].

In our study, more than half of the sample (57.6%) had depressive symptoms, presenting 25.4% of patients moderate (13.5%) or severe symptoms (11.9%). We also found high levels of anxiety (59.5% had level of SA and 49.2% of TA in the percentile range 75–99). In studies in which self-applied scales are used to assess anxiety and depression in women with CPP, high percentage of anxiety and depressive symptoms were found [9,55,56,57,58]. In one study that measured the intensity of anxiety and depression, they found moderate or severe anxiety in 40% and 13.1% respectively, and moderate-to-severe depression in 22.3% and 4.6% [9], similar to our findings; although we found a higher percentage of severe symptoms. In CPP-specific studies in which the frequencies of mental disorders were investigated with structured interviews, elevated frequencies of mood and anxiety disorders were found [59,60,61]. Some studies found also a high prevalence of somatization spectrum disorders [59,61] and substance abuse disorders [61], results we did not find. We found that anxiety and depression correlate positively and significantly (*p* = 0.74 with STAI-TA and *p* = 0.64 with STAI-SA), which is a correlation also found in a study analyzing the psychologic profile in women with CPP, although in that study it was no so relevant as we found (*p* = 0.575) [9].

It is known that depression is a risk factor for developing chronic pain and that chronic pain favors the development of depression [62,63]. Similar mechanisms are described for anxiety. Current theories about the association of chronic pain with depression and anxiety point to shared brain mechanisms that would explain the bidirectional relationship between chronic pain, anxiety and depression [63]. From a clinical point of view, it would be important to treat depression and anxiety in order to improve outcomes in these patients.

It is also important to note the correlation between the STAI-SA, STAI-TA and the BDI with the CPAQ questionnaire. This correlation was not clinically significant (r = −0.51, r = −0.52 and r = −0.51 respectively), but it is striking that most of the correlation is explained by the activity engagement (r = −0.59, r = −0.57 and r = −0.57, respectively) subscale and not by pain willingness (r = −0.16, r = −0.21 and r = −0.22, respectively). It would be important to study this difference in depth. The CPAQ questionnaire has been used in previous studies of chronic pain [64,65]. However, we have not found studies that analyze the subscales of this questionnaire in depth, usually focusing on the global outcome. These results could lead us to consider that, although work on acceptance of chronic pain is important and effective, emphasis should be placed on behavioral activation interventions given their apparent relationship with other types of psychopathology. This becomes even more important when we observe that almost half of our sample is not working (47.9%) because they are on temporary leave or unemployed. It would be interesting that future studies will examine the discriminative efficacy of both types of intervention, however it is usual for both to be presented together within a treatment plan [66].

Regarding personality traits, we found high mean levels of neuroticism, low extraversion and low conscientiousness in the NEO-FFI scale. Similar results were found on one paper that compared urologic chronic pelvic pain syndromes (UCPPS), other non-urologic chronic overlapping pain conditions (COPCs) and healthy controls (HC) in a mixed sample of males and females. They found that UCPPS and COPC participants reported higher neuroticism and lower extraversion than HC individuals, with medium effects sizes [7]. They also found that the low-extraversion group had more CPP symptom severity than those with high extraversion. In studies of pain management, low extraversion has been found to be highly correlated with depression, anxiety, as well as self-efficacy [14,15,67,68]. Another study that assessed the association of personality traits on treatment outcomes in male patients with chronic prostatitis/chronic pelvic pain syndrome (CP/CPP) found at baseline that patients with higher neuroticism may have greater levels of depression and somatization and low subjective feelings regarding their general health status, those with lower extraversion may have more CP/CPP symptoms and those with the low-conscientiousness group may have exhibited more depression [40]. No studies in females were found for comparison. Personality profiles have been studied in chronic pain in relation to the experience and ways of coping with pain. In a study by Soriano et al. [67] in chronic pain, they found that neuroticism, extraversion and, to a lesser extent, conscientiousness, are the primary dimensions, in terms of prediction, accounting for quality of life in patients with chronic pain. They also found that the group that scored very high on neuroticism, very low on extraversion, very low on openness, moderate on agreeableness and very low on conscientiousness showed more pain, poorer quality of life and less use of helpful coping strategies compared to the group with a different personality profile (high on neuroticism, moderate on extraversion, moderately low on openness, high on agreeableness, and moderate on conscientiousness). The personality profile we have found is similar to the first one described in this study.

With respect to the correlation between personality traits and other variables, we only found a significant and relevant positive correlation between the variable Neuroticism and trait anxiety (STAI-TA -*p* = 0.65) in our study. No other relevant and significant correlations were found between personality traits and depression, anxiety, catastrophizing or illness acceptance. They are studies that have found that neuroticism may contribute to greater catastrophizing about pain [14,36,69]. In our study, we found a statistically significative correlation between neuroticism and catastrophizing, although this correlation was not clinically relevant (*p* = 0.3783). Other descriptive studies in various chronic types of pain have found also high levels of neuroticism compared to control groups [14,15,16,17,18,19,20,21,22,23,24,25]. Several studies have focused on the study of neuroticism as this trait may affect the adjustment to ongoing pain, increasing suffering and disability, and potentially contributing to the development of chronic pain [14]. It could be interesting to study the relation between the personality profile as a cluster that we have found and the rest of variables, including the coping with pain.

With regard to catastrophizing, we found high levels in our sample (P25 = 13; P50 = 29; P75 = 38). The catastrophizing scale (PCS) ranges from 0 to 52 points, low punctuations being scarce catastrophizing and high punctuations elevated catastrophizing. In a previous study that analyzed pain catastrophizing in women with CPP, they found that 60 of 113 participants had a clinically relevant total score ≥ 30 in PCS [70]. In a meta-analysis analyzing psychological factors and pain catastrophizing in men with chronic prostatitis/chronic pelvic pain syndrome (CP/CPPS), they also found high catastrophizing [71]. Statistically significant positive correlations between pain catastrophizing scores and pelvic pain levels have been found in other studies in CPP [13,61,63,64]. Catastrophizing was also related in CPP patients with a poorer response to a variety of treatment strategies [13,57,63,65,66] and poorer quality of life [71]. Other studies have found positive correlations between catastrophizing, anxiety and depression [17,72,73,74,75,76], correlations we did not find in our study. These results seem to suggest that catastrophizing is a variable strongly associated with chronic pelvic pain, regardless of gender.

In light of our results and those of other studies in both women and men with CPP in which high catastrophizing is found, it would be important to include as part of the treatment psychotherapeutic strategies that could diminish catastrophizing in these patients.

The results obtained in pain acceptance (CPAQ) were similar to those found in fibromyalgia, with most of punctuations being below half of the maximum possible to obtained either in each subscale or in the total score. These results can be interpreted as low acceptance of pain. Pain acceptance is often a predictor of outcomes including quality of life among people with chronic pain attending pain clinics [77] and among people with fibromyalgia [78], chronic musculoskeletal pain [79] or chronic hemophilia-related joint pain [80]. To our knowledge, no specific study has assessed illness acceptance in CPP.

In summary, in addition to results similar to other studies in terms of anxiety, depression, catastrophism or neuroticism, we have found a specific personality profile and a low acceptance of pain not previously described in CPP.

In light of the results of our study and taking into account the multiple factors that influence CPP (anxiety, depression, catastrophizing, previous abuse, etc.), it becomes important to perform a complete psychological evaluation of patients in order to individualize and establish the best treatment for each patient. It is also important to consider what would be the better focus of treatment depending on the clinical state and how each patient is coping with illness. Given the importance of depression and anxiety in CPP, it should be a priority to determine whether patients need coadjutant psychopharmacological treatment for anxiety and/or depression as well as other specific treatments such as those for traumatic events in patients with a history of abuse, for substance abuse, etc. For this reason, in our program we offer patients a complete evaluation prior to psychotherapeutic group therapy in which we combine elements of cognitive-behavioral therapy, acceptance and commitment therapy and mindfulness. Through this technique, we intend to work through catastrophizing, illness acceptance, engagement in activities and management of anxiety in order to help to improve outcomes in patients with CPP. 

It would be interesting for future research to replicate our findings and include the pain diagnosis of patients in order to determine if different clusters of patients have different psychological characteristics. As previously mentioned, it would also be interesting to study the relation between the personality profile and the rest of the variables, including the coping with pain. Finally, it would be interesting if future research would study the usefulness of the group psychotherapeutic intervention for CPP.

The main limitations of our study are the small sample size and the lack of some of the data. This can lead to variations in the estimated frequencies of different variables. The unstructured interview may introduce an interrater variability as well as data loss that can introduce bias in the clinical diagnosis and loss of data. Pain diagnosis and other comorbid pain conditions should be registered for more complete information of patients. This study began with a clinical purpose, and we progressively adapted some of the questionnaires to our experience with patients with CPP until the approved protocol was developed. Because of this, there is a difference between the sample sizes of different scales, and it is difficult to carry out more complex statistical analyses. Our main objective was to describe the sample available to us. However, in the future, it would be important to have a larger sample size and a control group. This would allow us to investigate the relationships that the different variables described have with each other and with the CPP. 

## 5. Conclusions

Our study finds a personality profile and low pain acceptance not described previously in CPP. It also confirms data found in previous studies in which patients with CPP have higher levels of anxiety, depression, high neuroticism and high catastrophizing. Taking into account the importance of all these factors in outcomes in CPP, it is important to offer patients multidisciplinary treatment strategies that include a psychotherapeutic integrative approach that could help patients to manage and improve all the aspects mentioned.

## Figures and Tables

**Table 1 jcm-11-06345-t001:** Sociodemographic and personal clinical history variables.

Variable	N	%
Gender (N = 63)		
Women	63	100
Men	0	0
Marital status (N = 63)		
Single	15	23.81
Couples/Married	43	68.25
Divorced	5	7.94
Sons (N = 57)		
Childless	23	40.5
With children	34	59.65
Studies (N = 63)		
Secondary Education	8	12.7
Advanced Level	10	15.87
National Diploma	7	11.11
Higher National Diploma	11	17.46
Bachelor’s Degree	27	42.86
Work activity (N = 55)		
Student	2	3.17
Active worker	19	30.16
Temporary leave for work	15	27.27
Permanent incapacity for work	3	5.45
Unemployment	13	20.63
Retired	3	4.76
Personal psychiatric history (N = 56)		
No	15	26.79
Depression	12	21.43
Anxiety	8	14.29
Adaptive	7	12.5
Obsessive-compulsive	2	3.57
Psychosis	2	3.57
Other	10	17.85
Current psychopharmacological treatment (N = 43)		
Yes	19	44.19
Current treatment for pain (N = 50)		
Yes	40	80
Current substance use		
	Yes N (%)	No N (%)
Tobacco (N = 38)	9 (23.68)	29 (76.32)
Alcohol (N = 39)	14 (35.89)	25 (64.11)
Cannabis (N = 39)	2 (5.12)	37 (94.88)
Cocaine (N = 39)	1 (2.56)	38 (97.44)
Other toxics (N = 39)	0 (0)	39 (100)
History of abuse (N = 34)		
Yes	14	41.18
Family psychiatric history (N = 49)		
No	18	36.73
Depression	15	30.61
Anxiety	3	6.12
Substance use	4	8.16
Psychosis	4	8.16
Other	5	10.2

Abbreviation: N: Number of patients; %: Percentage.

**Table 2 jcm-11-06345-t002:** Means and percentiles of NEO-FFI, PCS and CPAQ scales.

Scale	Mean	Percentile		
NEO-FFI (N = 59) *		P25	P50	P75
Neuroticism	63.20 (H)	56 (H)	64 (VH)	71 (VH)
Extraversion	42.389 (L)	34 (VL)	42 (L)	52 (M)
Openness	48.54 (M)	39 (L)	47 (M)	57 (H)
Agreeableness	45.20 (M)	39 (L)	44 (L)	52 (M)
Conscientiousness	42.89 (L)	34 (VL)	41 (L)	51 (M)
PCS (N = 59)				
Rumination	9.16	5	11	13
Magnification	5.15	2	5	8
Helplessness	11.69	5	12	18
Total	25.84	13	29	38
CPAQ (N = 45)				
Activity engagement	31.12	23	30	39
Pain willingness	17.02	9	13	25
Total	48.22	38	49	58

Abbreviations: N: Number of patients; * Very low (VL 25–34; Low (L) 35–44; Medium (M) 45–54; High (H) 55–64 and Very high (VH) 65–75.

**Table 3 jcm-11-06345-t003:** BDI-FS scale: sample means and severity.

BDI-FS (N = 59)	Mean	N (%)			
		Minimal	Mild	Medium	Severe
	5.71 (Mild)	25 (42.38)	19 (32.20)	8 (13.56)	7 (11.86)

Abbreviations: N: Number of patients; %: Percentage.

**Table 4 jcm-11-06345-t004:** Sample means and percentages of the STAI questionnaire.

STAI	Mean	N (%) for Each Category of Percentiles				
		<50	50–75	75–85	85–99	>99
State Anxiety (N = 42)	31.35	9 (21.43)	8 (19.05)	7 (16.67)	16 (38.10)	2 (4.75)
Trait Anxiety (N = 59)	28.08	17 (28.81)	13 (22.03)	12 (20.34)	16 (27.12)	1 (1.7)

Abbreviations: N: Number of patients; %: Percentage.

**Table 5 jcm-11-06345-t005:** Pearson correlations between clinical scales.

		STAI		BDI-FS	CPAQ			PCS			
		SA	TA		AE	PW	T	R	M	H	T
STAI	SA	1									
	TA	0.74	1								
BDI-FS		**0.64 ***	**0.74 ***	1							
CPAQ	AE	**−0.59**	**−0.57**	**−0.57**	1						
	PW	−0.16	−0.21	−0.22	0.22	1					
	Total	−0.51	−0.52	−0.51	0.83	0.72	1				
PCS	R	0.22	0.45	0.37	−0.34	−0.21	−0.37	1			
	M	0.33	0.53	0.27	−0.21	−0.25	−0.29	0.67	1		
	H	0.33	0.54	0.45	−0.37	−0.34	**−0.45**	0.78	0.7	1	
	Total	0.30	**0.53**	0.40	−0.35	−0.33	−0.43	0.91	0.82	0.94	1

Abbreviations: Bold: statistically significant; *clinically relevant.

**Table 6 jcm-11-06345-t006:** Pearson correlations between NEO-FFI and clinical scales.

	NEUROTICISM-T	EXTRAVERSION-T	OPENNESS-T	AGREEABLENESS-T	CONSCIENTIOUSNESS-T
STAI-TA	**0.65 ***	**−0.53**	−0.14	−0.00	−0.05
STAI-SA	**0.54**	**−0.38**	**−0.31**	0.17	0.12
BDI-FS	**0.51**	**−0.48**	−0.26	0.06	−0.01
CPAQ-Tot	**−0.31**	**0.33**	**0.35**	−0.08	−0.17
CATASTROPHIZING-Tot	**0.38**	−0.19	−0.04	0.09	0.21

Abbreviations: Bold: statistically significant; *: clinically relevant.

## Data Availability

Data presented in this manuscript are available from the corresponding authors on reasonable request.

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
