# Peer review of "Psychological Profile in Women with Chronic Pelvic Pain"

_jcm, 2022, doi:10.3390/jcm11216345_

Round 1

Reviewer 1 Report

Because chronic pelvic pain (CPP) in women remains a burden for women suffering CPP and the health care professionals, sound research on the diagnostics and treatment modalities of this condition is urgently needed.

The authors of the manuscript ‘Psychological profile in women with Chronic Pelvic Pain’ report about the findings of a socio-demographic and psychological assessment of 63 women with CPP.

After having read this manuscript I am feeling a bit disappointed. Some important question remain unanswered : what are the new results based on this research considering existing knowledge? And: what’s the main research question of the current study and what could be the secondary questions? Because these issues are not addressed explicitly, the manuscript remains difficult to read and therefore not systematically constructed.

Some specific remarks

Introduction

It seems that the authors want to focus on ‘personality traits ‘associated with CPP. However no (research) question was formulated, only a very broad purpose of the study is formulated. ( line 69-70). It is important that the authors focus more about what they intend (have intended) to study and about what the arguments are to do so.

Materials and Methods

It remains unclear what the exact procedure was for inclusion of the participants. I understand that the presented data are a part of another study?. What was the purpose of this other study?

How were the potential participants informed about the study? What were the inclusion criteria ?did the potential participants have to have pain for one or more days per month? What was the population from where the participants were recruited? For now, the study population seems a selected and biased group of women with CPP. The authors have not addressed this issue.

How many women refused a referral to the psychiatric department, although they suffered pain? How did the authors get informed consent of the participants, at what moment of the inclusion procedure? ? Did they get any (financial) compensation?

The clinical interview by two persons, was used to collect socio demographic data as well as relevant clinical data. Because it is unclear whether this interview is structured, semi- structured or unstructured, I wonder whether the results are not biased by an interrater variability, specifically regarding current pain, activities that affect pain severity, coping with pain and also regarding a history of sexual, emotional and physical abuse. What is said in the Lines 91-93 increase my worries.

Also were various mentioned diagnosis checked with medical record information or were the facts based on the memory of the participant ( for instance about depression, anxiety disorder in the past...etc)

Data analyses

Because no clear hypothesis was formulated and also the data set is incomplete ( various missing values) only descriptive and correlational statistics have been used. The correlational analysis seem to be performed without a clear hypothesis.

As I understand it well, this study is an uncontrolled study. As a comparison data of a study with patients with fibromyalgia and endometriosis was used; No control group of women without chronic pelvic pain, was used to compare the results.

Results:

Table 1 and table 2 have to be restructured to one table, the so called TABLE 1 ( all baseline sociodemographic and psychological determinants in one table). Only important results have to mentioned in the text and not all those percentages that are logic ( lines 140-144)

Table 2 contains ‘current treatment’ yes or no ( the ‘no’ can be deleted)

But, do the authors know which kind of treatment these women have (had)? For instance: medication, relaxation training, cognitive behavioral therapy, massage, acupuncture and so on..

How important is it to know what the different products are that can be used if associated with substance use.

Emotional and sexual abuse are investigated, but what is known about physical abuse? And why not using ‘one factor like ‘abuse in the past’” because it is really difficult to make a clear distinction between physical and emotional abuse.

Why is it important to know about the family psychiatric history?

It is not relevant to mention that two men were referred for evaluation, (line 138-139)

Clinical scales results

I don’t understand the reason why the authors present the NEO FFI data ranges. Are these ranges defined and according to the manual of the questionnaire? What is the clinical meaning of these results? A similar question can be asked regarding the presentation of the results of the BDI FS and STAI.

What is the meaning of the correlational analyses?

Conclusion

Based on all these above mentioned issues and questions, it seems not worthwhile to address the discussion. The authors have to restructure their manuscript in such a way that the goal of the study is clear based on current knowledge.

unfortunately doe not add specific new information

Author Response

Dear Mr/Ms. Reviewer:

Thank you very much for taking the time to review our article and giving us your feedback. We have thoroughly reviewed all the critical remarks and suggestions for the improvement of the paper. Please find attached below the modifications of the article based on the questions you have asked us. In the same way, we attach the final article with the changes in red so that you can see how the proposed changes were integrated.

Introduction

It seems that the authors want to focus on ‘personality traits ‘associated with CPP. However no (research) question was formulated, only a very broad purpose of the study is formulated. (line 69-70). It is important that the authors focus more about what they intend (have intended) to study and about what the arguments are to do so.

As we explained in the introduction, there are studies on the importance of psychological and personality variables in chronic pain. However, not much studies have studied these variables in chronic pelvic pain. For this reason, our aim has been to describe the psychological variables of our clinical sample. We hope to find a psychological profile similar to that found in previous studies in CPP and with other types of chronic pain. Please find attached the extended and corrected text based on your recommendation:

(Lines 78-89) “The main purpose of this study is to characterize the baseline psychological characteristics of women treated in a multidisciplinary group of CPP.

Given the importance that psychological and personality variables have been found to have in chronic pain in general and in CPP in particular [2,5-24,26-35], the aim of this work is to investigate whether these variables are also present in the CPP patients of our study and to expand, if possible, the knowledge we have about the psychological profile of these patients.

Our main hypothesis is that the study participants will have high levels of depression, anxiety and catastrophizing, low acceptance of pain and high levels of neuroticism as part of the personality dimension. As a secondary objective, we intend to investigate whether the psychological and personal variables are related to each other with the hypotheses that anxiety, depression and catastrophizing may be positively correlated.”

Materials and Methods

It remains unclear what the exact procedure was for inclusion of the participants. I understand that the presented data are a part of another study?. What was the purpose of this other study?

How were the potential participants informed about the study? What were the inclusion criteria? did the potential participants have to have pain for one or more days per month? What was the population from where the participants were recruited? For now, the study population seems a selected and biased group of women with CPP. The authors have not addressed this issue.

How many women refused a referral to the psychiatric department, although they suffered pain? How did the authors get informed consent of the participants, at what moment of the inclusion procedure? Did they get any (financial) compensation?

Participants were diagnosed with chronic pelvic pain by their referring physician. This pain, by definition, as explained in the introduction, has to be present for more than 6 months. The pain can be continuous or intermittent depending on its origin. The medical professionals from different services that treated and diagnosed women with CPP offered them referral for evaluation for the psychotherapy group when they saw that the pain had a significant impact on their daily functioning due to its association with high emotional distress and/or psychopathology.

We do not have data on patients who were offered referral and did not accept it. We have data on those who, once referred, didn´t attend the programed visit or do not participate in the study. Ninety patients were referred for evaluation. Of these, five did not attend the scheduled appointment and twenty-one didn´t participate in the research.

In summary, and answering your questions, the new reworded text in terms of participants is as follows (lines 97-108 that replace lines 77-82 of previous manuscript):

“The criteria the deriving physicians used for referral to psychiatry evaluation were emotional distress and/or the presence of psychopathology associated with CPP that interfered with its outcome. All patients who agreed to fill in the questionnaires and that these could be used for research purposes were included. The verbal informed consent was obtained at the end of the evaluation visit. Participants didn´t receive any financial compensation. Ninety patients were referred for evaluation. Of these, five did not attend the scheduled appointment and twenty-one didn´t participate in the research. However, non-participation in the research did not exclude the possibility of joining the psychotherapy group.

This research is part of a larger project that aims to assess the effectiveness of the psychotherapy group in CPP. The project of which this study is a part was approved by the Drug Research Ethics Committee”

The clinical interview by two persons, was used to collect socio demographic data as well as relevant clinical data. Because it is unclear whether this interview is structured, semi- structured or unstructured, I wonder whether the results are not biased by an interrater variability, specifically regarding current pain, activities that affect pain severity, coping with pain and also regarding a history of sexual, emotional and physical abuse. What is said in the Lines 91-93 increase my worries.

Also were various mentioned diagnosis checked with medical record information or were the facts based on the memory of the participant (for instance about depression, anxiety disorder in the past...etc)

               The clinical interview was unstructured. Although the clinicians had a script to cover all the relevant information that we wanted to asses, this was not always possible, which explains the loss of some data. This may introduce interrater variability that should be noted as a limitation in the study limitations section.               The patient's electronic medical record was consulted before the first interview to collect as much clinical information as possible. We have access to the primary care record, our hospital records and discharge reports of other hospital from our region. Despite all this, it was not possible to have 100% of the information based on rigorous medical records and part of the information about past clinical history was based on the memory of the participant.                The lines 91-93 are confusing and inexact. The paragraph will be rewrite.  

The revised text is as follows:

(Lines 113-116)

 “An unstructured interview was carried out to collect socio-biographical data as well as the most relevant clinical history. Before the clinical interview, the patient's medical history was reviewed in order to obtain the most precise history as possible”

(Lines 119-120 replace lines 91-93 of previous manuscript)

“Although the evaluators had a script to cover all the relevant information that we wanted to asses, this was not always possible, which explains the loss of some data”.

(Lines 396-400)

“…The unstructured interview may introduce an interrater variability as well as data loss. Pain diagnosis and other comorbid pain conditions should be registered for more complete information of patients This study began with a clinical purpose, and we progressively adapted some of the questionnaires to our experience with patients with CPP until the approved protocol was developed…”

Data analyses

Because no clear hypothesis was formulated and also the data set is incomplete (various missing values) only descriptive and correlational statistics have been used. The correlational analysis seem to be performed without a clear hypothesis.

As I understand it well, this study is an uncontrolled study. As a comparison data of a study with patients with fibromyalgia and endometriosis was used; No control group of women without chronic pelvic pain, was used to compare the results.

Modification (line 92) “The study is a cross-sectional descriptive study without a comparison group”

The Chronic Pain Acceptance Questionnaire (CPAQ) was validated in Spain in patients with fibromyalgia and the Pain Catastrophizing Scale (PCS) in patients with endometriosis. These scales have not been validated in our country in general CPP. This is the reason why we compare our results with the validated versions in our country.

Hypothesis were added (lines 85-89):

“…Our main hypothesis is that the study participants will have high levels of depression, anxiety and catastrophizing, low acceptance of pain and high levels of neuroticism as part of the personality dimension. As a secondary objective, we intend to investigate whether the psychological and personal variables are related to each other with the hypotheses that anxiety, depression and catastrophizing may be positively correlated”

Results:

Table 1 and table 2 have to be restructured to one table, the so called TABLE 1 (all baseline sociodemographic and psychological determinants in one table). Only important results have to mentioned in the text and not all those percentages that are logic (lines 140-144)

Table 2 contains ‘current treatment’ yes or no (the ‘no’ can be deleted)

The indicated changes have been applied. You can see them in the modified version of the article

(Line 234-248) “Table 1. Socio-demographic and Personal clinical history variables”

But, do the authors know which kind of treatment these women have (had)? For instance: medication, relaxation training, cognitive behavioral therapy, massage, acupuncture and so on…

The type of treatment the patients have had and currently have, have been asked in the unstructured interview but the different treatments were not included in the analysis, we only registered it as yes or no. It would be interesting for future research to include more precisely this kind of information.

How important is it to know what the different products are that can be used if associated with substance use.

We registered different substances in order to know whether patients consume more stimulant (i.e., cocaine) o more sedative substances that could contribute to relief pain (i.e., cannabis, alcohol or heroine).

Emotional and sexual abuse are investigated, but what is known about physical abuse? And why not using ‘one factor like ‘abuse in the past’” because it is really difficult to make a clear distinction between physical and emotional abuse.

As you have suggested only one factor of abuse was included (see Table 1 in line 224 of the corrected text)

Why is it important to know about the family psychiatric history?

Family psychiatric history can indicate more personal vulnerability to psychiatric disorders

It is not relevant to mention that two men were referred for evaluation, (line 138-139)

You right, we have eliminated it from the text

Clinical scales results

 I don’t understand the reason why the authors present the NEO FFI data ranges. Are these ranges defined and according to the manual of the questionnaire? What is the clinical meaning of these results? A similar question can be asked regarding the presentation of the results of the BDI FS and STAI.

All the results provided by the NEO-FFI, BDI FS and STAI questionnaires are in accordance with the official scales for the Spanish population in the questionnaire application manuals. The official references to which we refer can be found in the bibliography of the article. 

Regarding the question about the score ranges (high, medium, low...), in our opinion it is fundamental for the understanding of the article to include the official ranges of the questionnaires used. The numerical scores obtained in the questionnaires do not provide the reader with sufficient information about the clinical importance of the score. That is, they do not say what it means to have obtained that score. It is therefore necessary to indicate in which range the score falls in order to be able to give clinical meaning to the data obtained.

What is the meaning of the correlational analyses?

In this study, the analysis of correlation is appropriate because it is not possible to establish causality between variables (this is not an experimental study but a descriptive one). However, it is possible to describe the relationship between variables, for example, between the different scales.  

Reviewer 2 Report

A well-written article that tackles a difficult topic. I have no comments. It would have been more interesting if men with CPS were enrolled but even so, it bring some light in the pshychological profile of women with CPS.

Minor comment: please insert, if possible, the underlying gynecological condition  of the women enrroled.

Author Response

Comment to Authors

A well-written article that tackles a difficult topic. I have no comments. It would have been more interesting if men with CPS were enrolled but even so, it bring some light in the pshychological profile of women with CPS.

We also think that it would be interesting to have men in our sample but unfortunately almost no men were derived to our program. Recently the urology department was invited to participate in the multidisciplinary group of chronic pelvic pain so it is possible that in the future also men could be included in the program.

Minor comment: please insert, if possible, the underlying gynecological condition  of the women enrroled”

This is a very pertinent comment but we don´t have in our database this information at present. In future research we have considered including this point.

Reviewer 3 Report

Chronic Pelvic Pain Syndrome (CPP) is a set of symptoms that mainly present with localized pain in the pelvic or perineal area and that can radiate to the lumbar region, groin, vagina and vulva, the suprapubic region, to the sacrum-coccyx and the root of the thighs. For it to be considered as such it must last at least 6 months and it does not present pathogenetic mechanisms typical of acute pain.

The aim of this study is to define the baseline psychological characteristics of women treated in a multidisciplinary group of CPP.

Comment to Authors

Authors should be congratulated for the interesting topic discussed.

The manuscript is easily readable, well-written, and tables are clearly described, but it is lacking in several points that would add value to the entire manuscript:

-        Please, provide the aim of the study in the abstract too.

-        Please, authors should use more up-to-date references.

-        Please, the authors should describe in more detail the characteristics of the CPP and in what context of conditions it is embedded. In addition, it would be important to include any other pathological conditions that may accompany the pathology. In this article, all these requests are exhaustively described. Reading is suggested (https://doi.org/10.1111/jop.13097). It would increase the scientific resonance of the paper.

-        Materials and methods are robust.

Author Response

Comment to Authors

Authors should be congratulated for the interesting topic discussed.

The manuscript is easily readable, well-written, and tables are clearly described, but it is lacking in several points that would add value to the entire manuscript:

-        Please, provide the aim of the study in the abstract too.

      It was included in the abstract

(Line 19-20) “…The main purpose of this study is to characterize the baseline psychological characteristics of women with CPP…”

-        Please, authors should use more up-to-date references.

      Some up to date references were added (references 5,7,8,58)

-        Please, the authors should describe in more detail the characteristics of the CPP and in what context of conditions it is embedded. In addition, it would be important to include any other pathological conditions that may accompany the pathology. In this article, all these requests are exhaustively described. Reading is suggested (https://doi.org/10.1111/jop.13097). It would increase the scientific resonance of the paper.

We have added some more information about CPP and its association with other pain syndromes. We do not have easily accessible information about our patients' pain diagnoses and comorbid pain conditions for the present paper. For future research it would be important to include this information. This fact was included in the limitations section.

The text changes appear in red in the revised version and they are as follows:

(Lines 42-51): “…It can arise from the gynecologic, urologic, gastrointestinal, musculoskeletal and neuro-vascular systems and many factors contribute to its development. The most common gynecological causes of CPP are endometriosis, pelvic inflammatory disease and adhesions, the most common digestive cause is irritable bowel syndrome and the most frequent urologic causes are chronic prostatitis and interstitial cystitis [5, 6]. The association of CPP with pain conditions in other location is not infrequent [5,7,8]. In no urologic CPP it was found that 40% of patients had multiple pain syndromes [7] and in urologic CPP a 38% were associated to other pain syndromes [5] Fibromyalgia, irritable bowel syndrome and chronic fatigue syndrome were those more frequently associated with CPP [5], [7]…”

(Line 395-400) “… The main limitations of our study are the small sample size and the lack of some of data. The unstructured interview may introduce an interrater variability as well as data loss. Pain diagnosis and other comorbid pain conditions should be registered for more complete information of patients This study began with a clinical purpose, and we progressively adapted some of the questionnaires to our experience with patients with CPP until the approved protocol was developed.”

-        Materials and methods are robust.

Round 2

Reviewer 1 Report

The authors have answered most of my questions properly and were able to  improve their manuscript a lot. However, the quality of the manuscript remains  low. I cannot agree with publication of this last version.

Two main problems still exist:

First: (lines 78-89) what is the main aim/purpose of this study and what are the secondary goals? And why would the study population be a specific and/or other group of patients than what's already known from literature regarding psychological characteristics? What is new or what would the findings of this study add to the current knowledge?

Secondly: the discussion has to be restructured thoroughly. The present discussion is difficult to read because lack of a clear structure

Start with a summary of the key results with reference to the study objectives and study hypothesis.

Give an interpretation of the results , also in comparison with results from similar studies. Note that a comparison of own findings with study results in male study participants is seriously hampered by gender differences  ( for instances line 374-375).

Discuss  the limitations of the study and its  consequences for the validity of the study results and not only that there are limitations ( lines  406-416)

What are  the clinical implications of the findings?

What are suggestions for further research?

Reviewer 3 Report

Authors answered all comments and suggestions.
